# Atg9 antagonizes TOR signaling to regulate intestinal cell growth and epithelial homeostasis in *Drosophila*

Jung-Kun Wen[1,2], Yi-Ting Wang[1,3], Chih-Chiang Chan[4], Cheng-Wen Hsieh[4], Hsiao-Man Liao[1], Chin-Chun Hung[1], Guang-Chao Chen[1,2,3]*

[1]Institute of Biological Chemistry, Academia Sinica, Taipei, Taiwan; [2]Genome and Systems Biology Program, College of Life Science, National Taiwan University, Taipei, Taiwan; [3]Institute of Biochemical Sciences, College of Life Science, National Taiwan University, Taipei, Taiwan; [4]Graduate Institute of Physiology, National Taiwan University College of Medicine, Taipei, Taiwan

**Abstract** Autophagy is essential for maintaining cellular homeostasis and survival under various stress conditions. Autophagy-related gene 9 (Atg9) encodes a multipass transmembrane protein thought to act as a membrane carrier for forming autophagosomes. However, the molecular regulation and physiological importance of Atg9 in animal development remain largely unclear. Here, we generated *Atg9* null mutant flies and found that loss of *Atg9* led to shortened lifespan, locomotor defects, and increased susceptibility to stress. *Atg9* loss also resulted in aberrant adult midgut morphology with dramatically enlarged enterocytes. Interestingly, inhibiting the TOR signaling pathway rescued the midgut defects of the *Atg9* mutants. In addition, Atg9 interacted with PALS1-associated tight junction protein (Patj), which associates with TSC2 to regulate TOR activity. Depletion of *Atg9* caused a marked decrease in TSC2 levels. Our findings revealed an antagonistic relationship between Atg9 and TOR signaling in the regulation of cell growth and tissue homeostasis.
DOI: https://doi.org/10.7554/eLife.29338.001

*For correspondence: gcchen@gate.sinica.edu.tw

Competing interests: The authors declare that no competing interests exist.

## Introduction

Autophagy is a highly regulated lysosomal degradation process by which intracellular components are degraded and recycled for cell viability and homeostasis. There is increasing evidence of the importance of autophagy in a variety of physiological and pathological processes, including differentiation, development, aging, and tumorigenesis (*Jiang and Mizushima, 2014*; *Mizushima and Levine, 2010*). The autophagic pathway is controlled by a series of evolutionarily conserved autophagy-related (Atg) proteins to generate double-membraned vesicles, the autophagosomes, which subsequently fuse with lysosomes for the degradation of their contents (*Feng et al., 2014*). Atg9 is the only multi-spanning transmembrane protein in the family and is involved in promoting lipid transport to autophagosomes during their formation (*Tooze, 2010*; *Yamamoto et al., 2012*). In yeast, distinct from other Atg proteins, Atg9 cycles between cytoplasmic membrane pools and the preautophagosomal structure (PAS) (*Mari et al., 2010*). The Atg9-containing vesicles are recruited to PAS by the Atg1-kinase complex during autophagosome formation (*Rao et al., 2016*; *Suzuki et al., 2015*). Similarly, mammalian Atg9 (mAtg9) localizes to the trans-Golgi network, endosomal system, and plasma membrane under normal conditions, whereas it translocates to autophagic membranes upon autophagy induction (*Orsi et al., 2012*; *Popovic and Dikic, 2014*; *Puri et al., 2013*; *Young et al., 2006*). The trafficking of mAtg9 is important for autophagy induction, and several proteins, including Ulk1, ZIPK, p38IP, TRAPPC8, TBC1D5, and the AP2 complex (*Imai et al., 2016*; *Lamb et al., 2016*;

*Popovic and Dikic, 2014*; *Tang et al., 2011*; *Webber and Tooze, 2010*; *Young et al., 2006*), regulate the spatio-temporal distribution of mAtg9 during autophagy.

In addition to autophagy, mAtg9 can modulate dsDNA-induced innate immune responses by regulating the STING-TBK1assembly (*Saitoh et al., 2009*). Recently, Imagawa et al showed that mAtg9 also plays a role in necrotic programmed cell death during bone morphogenesis (*Imagawa et al., 2016*). Our previous study in *Drosophila* showed that Atg9 functions not only as an essential component of autophagy, but also interacts with *Drosophila* tumor necrosis factor receptor-associated factor 2 (dTRAF2) to regulate ROS-induced c-Jun N-terminal kinase (JNK) signaling, including JNK-mediated autophagy activation and intestinal stem cell (ISC) proliferation (*Tang et al., 2013*). Moreover, oxidative stress-induced autophagy can inhibit JNK activity through a negative feedback mechanism to prevent the over-activation of JNK-mediated stress responses, thereby helping the maintenance of midgut homeostasis. However, the molecular regulation and physiological function of Atg9 remain largely unknown.

Target of rapamycin (TOR), a serine/threonine kinase, functions as a central player in the regulation of cell growth and metabolism in response to various environmental stimuli, including nutrient status, growth factors, and amino acids (*Saxton and Sabatini, 2017*). Under nutrient-rich conditions, TOR promotes protein synthesis and energy metabolism while suppressing autophagy (*Russell et al., 2014*). Under nutrient deprivation conditions, TOR is inhibited, leading to the induction of autophagy. TOR negatively regulates autophagy by phosphorylating and inhibiting Atg1/Unc51-like kinase 1 (Ulk1) complex activity (*Alers et al., 2012*). The Atg1/Ulk1 kinase is thought to act as the most upstream autophagy regulator for the initiation of autophagosome formation (*Itakura and Mizushima, 2010*). Atg1/Ulk1 recruits downstream Atg proteins to the phagophore assembly site and phosphorylates several Atg proteins, including the Ambra1/Beclin1/Vps34 complex and Atg9 (*Cheong et al., 2008*; *Di Bartolomeo et al., 2010*; *Papinski et al., 2014*; *Russell et al., 2013*). Interestingly, recent studies have shown that Atg1/Ulk1 can negatively regulate TOR signaling in *Drosophila* and mammals (*Lee et al., 2007*; *Scott et al., 2007*), suggesting a tight interplay between Atg1/Ulk1-dependent autophagy and TOR-mediated cell growth.

Here, we generated null mutants for *Drosophila* Atg9, and showed that loss of *Atg9* severely impairs starvation-induced and developmental autophagy. *Atg9* null mutant flies exhibited dramatically reduced lifespans, climbing defects, and hypersensitivity to stress. Surprisingly, ablation of *Atg9* also caused increased TOR activity and aberrant enlargement of intestinal epithelial cells in the adult *Drosophila* midgut. Similar intestinal defects were observed in *Atg1, Atg13* and *Atg17/Fip200* depletion mutants. We further identified PALS1-associated tight junction protein (Patj) as an Atg9-interacting protein. In mammals, the polarity protein Patj interacts with tuberous sclerosis complex 2 (TSC2), a negative regulator of TOR signaling, to regulate TOR activity (*Massey-Harroche et al., 2007*). Strikingly, overexpression of *Patj* and *TSC1-TSC2* suppressed adult midgut defects of *Atg9* mutants. Depletion of *Atg9* resulted in a dramatic decrease in TSC2 levels. Our findings revealed a novel negative feedback loop by which Atg9 inhibits TOR signaling to regulate cell growth and tissue homeostasis.

## Results

### Generation of *Drosophila* Atg9 mutant fly

Our previous studies showed that *Drosophila* Atg9 interacts with dTRAF2 to regulate JNK activation, autophagy induction, and midgut homeostasis under oxidative stress conditions (*Tang et al., 2013*). To investigate the physiological and developmental functions of Atg9, we generated *Atg9* null mutants using two different approaches. First, we replaced the *Atg9* open reading frame with a Gal4 knock-in cassette (*Atg9$^{Gal4KO}$*) using the ends-out homologous recombination approach (*Figure 1A*) (*Chan et al., 2011*). The Gal4 knock-in can be used for gene expression under *Atg9* endogenous regulatory elements in the *Atg9* mutant background. Second, we employed the CRISPR/Cas9 gene editing approach to replace a short coding region in the first exon of *Atg9* with the attPX-3-frameStop-floxed 3xP3-RFP cassette (*Kondo and Ueda, 2013*), which leads to a prematurely truncated *Atg9* mutant (*Atg9$^{d51}$*) (*Figure 1A*). The homozygous *Atg9$^{Gal4KO}$* and *Atg9$^{d51}$* flies and trans-heterozygous *Atg9$^{Gal4KO}$/Atg9$^{d51}$* flies are semi-lethal, with a few escapers. Interestingly, the escapers produce no offspring, suggesting fertility defects in *Atg9* mutants. We next

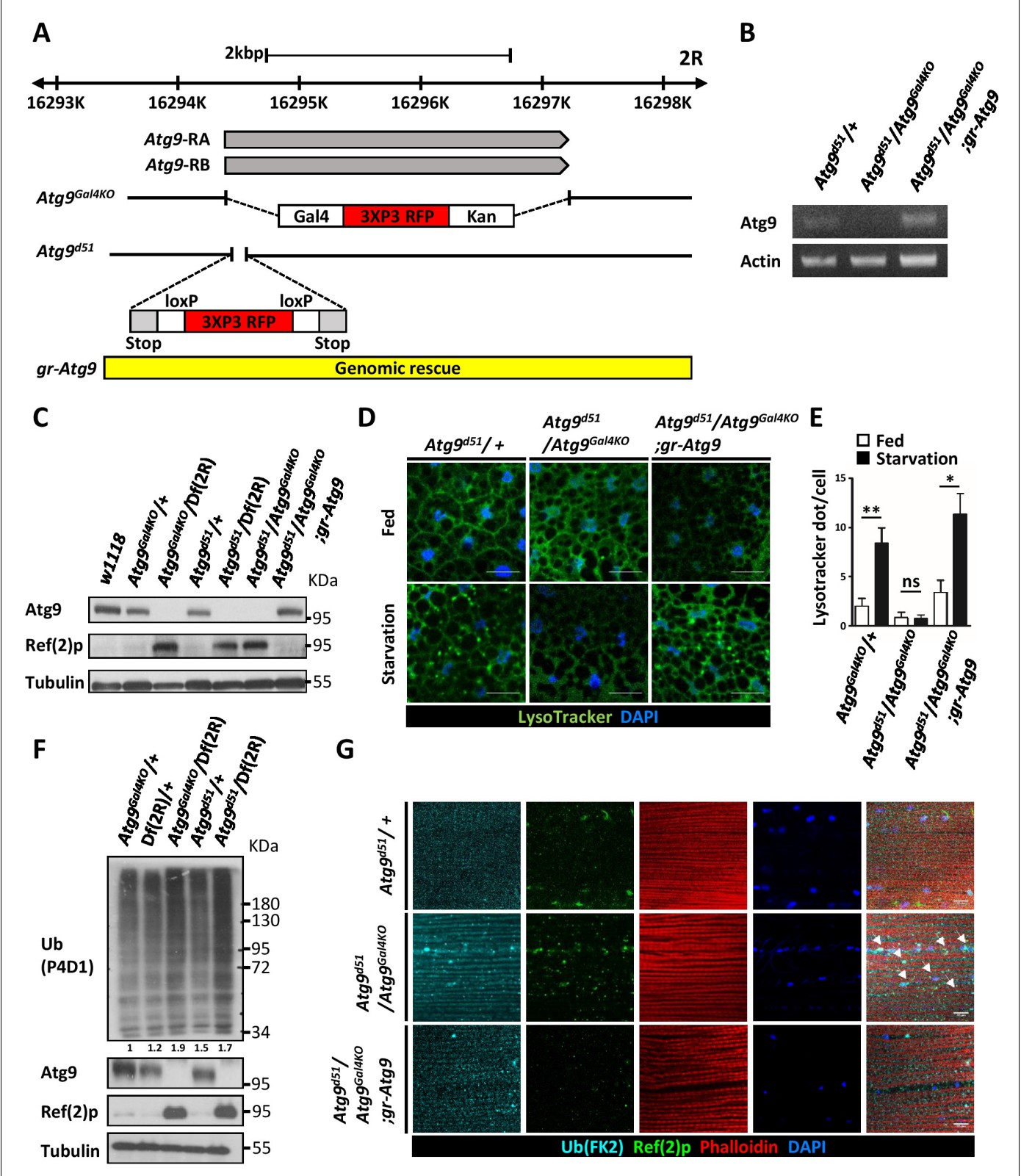

**Figure 1.** Generation of mutations in *Drosophila Atg9*. (**A**) Schematic view of *Atg9^Gal4KO* and *Atg9^d51* mutations relative to the *Atg9* transcripts. For the *Atg9^Gal4KO* mutation, the complete *Atg9* open reading frame was replaced with a Gal4 knock-in cassette. For *Atg9^d51* mutation, the 52–102 bp after the *Atg9* start codon was replaced with the attPX-3-frameStop-floxed 3xP3-RFP cassette. (**B**) RT-PCR analysis of *Atg9* mRNA expression level in control, mutant and *Atg9* genomic rescue adult flies. *Atg9* mRNA levels were undetectable in the *Atg9* mutant. (**C**) Western blots show the endogenous Atg9

*Figure 1 continued on next page*

*Figure 1 continued*

protein in control and *Atg9* genomic rescue flies but fail to detect the protein in mutants. (D) LysoTracker Green staining reveals that starvation-induced autophagy is strongly reduced in *Atg9* mutant fat bodies, compared with controls. Scale bar: 5 µm. (E) Quantification of data shown in (D). n ≥ 10, data are mean ±s.e.m. *p<0.05, **p<0.01. ns, not statistically significant. (F) Western blots show markedly increased Ref(2)P and ubiquitinated protein levels in *Atg9* mutants. (G) Immunostaining of *Drosophila* thoracic muscles with anti-Ub (FK2) and anti-Ref(2)p antibodies showed an accumulation and colocalization of polyubiquitin protein aggregates and Ref(2)p (arrowheads) in *Atg9* mutant flies. Scale bar: 10 µm. *Df* refers to *Df(2R)Exel7142*, which removes *Atg9* and flanking genes.

DOI: https://doi.org/10.7554/eLife.29338.002

The following source data and figure supplements are available for figure 1:

**Source data 1.** Quantification of lysotracker dots.

DOI: https://doi.org/10.7554/eLife.29338.004

**Figure supplement 1.** Loss of *Atg9* leads to impaired developmental autophagy and delayed degradation of larval midgut.

DOI: https://doi.org/10.7554/eLife.29338.003

**Figure supplement 1—source data 1.** Quantification of lysotracker dots and gastric caeca size.

DOI: https://doi.org/10.7554/eLife.29338.005

compared Atg9 expression in wild-type and mutant flies. We confirmed the lack of Atg9 expression in the mutants by RT-PCR and Western blot analysis (*Figure 1B and C*). Importantly, the gene expression and semi-lethality of *Atg9* mutants can be fully rescued by a 5.8 kb genomic construct encompassing the *Atg9* transcript and its endogenous regulatory regions (*Figure 1A–C*). These results demonstrated that $Atg9^{Gal4KO}$ and $Atg9^{d51}$ specifically disrupt Atg9 function and act as null mutants.

## *Atg9* mutants have impaired autophagy and increased ubiquitination

RNAi-mediated knockdown of *Atg9* inhibits starvation-induced autophagy and developmental autophagy in the larval fat body (*Bader et al., 2015*; *Tang et al., 2013*). To determine whether the newly generated *Atg9* null mutants also exhibit autophagic defecs, we first stained the larval fat body with the pH-sensitive fluorescent dye LysoTracker, which has been widely used to detect acidic lysosomes and autolysosomes. The LysoTracker Green staining was faint and diffuse in well fed control animals, whereas nutrient deprivation resulted in a strong punctate LysoTracker Green staining (*Figure 1D*). Notably, loss of *Atg9* dramatically blocked the starvation-induced punctate staining, compared to the controls (*Figure 1E*). Because inhibiting autophagic activity often results in the accumulation of autophagic substrate p62/SQSTM1 and ubiquitinated protein aggregates (*Komatsu and Ichimura, 2010*), we investigated the effect of Atg9 on protein ubiquitination and autophagic degradation of Ref(2)P, the *Drosophila* homolog of p62. Compared to the control, *Atg9* mutants had a dramatic increase in Ref(2)p levels and ubiquitin aggregate formation (*Figure 1F and G*). Moreover, *Atg9* null mutants displayed impaired developmental autophagy in the larval fat body and larval midgut (*Figure 1—figure supplement 1A and B*). These results together demonstrate the essential role of Atg9 in autophagy during development and in response to starvation.

## *Atg9* mutants exhibit shortened lifespan, locomotor defects, and increased susceptibility to stresses

Many *Drosophila* autophagy mutants display reduced lifespan and decreased climbing activity (*Juhász et al., 2007*; *Kim et al., 2013*). To gain more insight in the physiological function of Atg9, we analyzed the effect of *Atg9* gene ablation on the lifespan of *Drosophila*. As shown in *Figure 2A*, we found that the lifespan of *Atg9* mutants was greatly reduced compared with that of *Atg9* heterozygous control flies (48% decrease in male mean lifespan and 53% in female, p<0.001) or *Atg7* mutants (38% decrease in male mean lifespan and 40% in female, p<0.001) under normal conditions. The negative geotaxis assay also revealed that *Atg9* mutants exhibited a significantly lowered climbing activity than that of the control flies (*Figure 2B*). The locomotion defects of *Atg9* mutants were substantially suppressed by expressing the *Atg9* genomic rescue construct, suggesting that the mobility defects were indeed caused by interruption of *Atg9* gene expression. Moreover, loss of *Atg9* leads to dramatically decreased viability under starvation and oxidative stress conditions (*Figure 2C,D*). We thus conclude that, like other autophagy mutants, Atg9 also regulates *Drosophila* lifespan, mobility, and susceptibility to stresses.

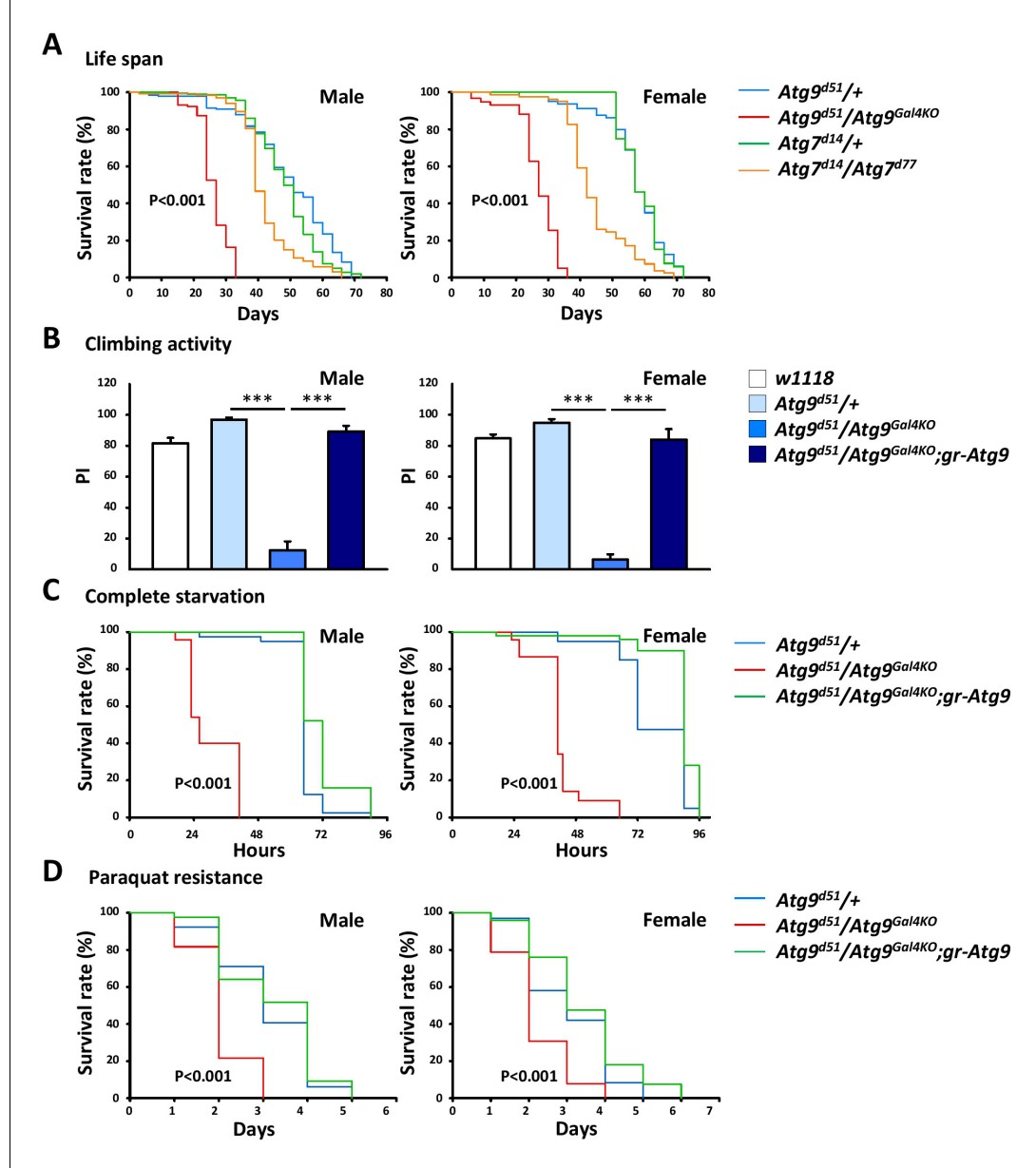

**Figure 2.** The *Atg9* mutant flies display shortened lifespan, locomotor defects and decreased stress tolerance. (**A**) Both *Atg7* and *Atg9* mutant flies showed shortened lifespan compared with control (Log-rank test, p<0.001). Comparing with *Atg7* mutant, *Atg9* mutant flies exhibited dramatically shortened lifespan. (**B**) Climbing analysis showed that *Atg9* mutant exhibit decreased climbing activity in both male and female flies, compared with controls (***p<0.001). The locomotion defects can be rescued by expressing the *Atg9* genomic rescue construct. (**C–D**) Both male and female *Atg9* mutant flies die faster than control flies under complete starvation conditions (Log-rank test, p<0.001) or on paraquat treatment (Log-rank test, p<0.001).

DOI: https://doi.org/10.7554/eLife.29338.006

The following source data is available for figure 2:

**Source data 1.** Survival rate and climbing activity of control and *Atg9* mutants.

DOI: https://doi.org/10.7554/eLife.29338.007

## Atg9 is required for proper adult midgut morphogenesis

The *Drosophila* adult midgut shares many similarities with the mammalian intestine and has emerged as an attractive model system to study stem cell proliferation and differentiation (*Jiang and Edgar, 2012*). Our previous study showed that *Drosophila* Atg9 is involved in regulating adult midgut

homeostasis upon ROS stimulation and bacterial infection (*Tang et al., 2013*). To examine whether Atg9 has a function in maintaining intestinal homeostasis under normal conditions, we examined the adult midgut morphology of the *Atg9* mutant. The *Drosophila* adult midgut consists of a tubular, monolayered epithelium surrounded by visceral muscles (*Micchelli and Perrimon, 2006*). In striking contrast, the *Atg9* mutant adult midgut is markedly shortened and significantly thickened in the posterior region, compared to control flies (*Figure 3A*). Moreover, phalloidin staining of actin filaments in the visceral muscles revealed a severe disruption in the continuity of the visceral mesoderm layer surrounding the gut (*Figure 3B*). To assess whether the loss of *Atg9* would affect intestinal barrier function, we examined the intestinal integrity by feeding flies of different ages with a nonabsorbable blue food dye (*Rera et al., 2011*). As expected, the dye was restricted to the digestive tract in young control flies (10 days, Smurf- fly), whereas the dye was seen throughout the body in approximately 7% of the aged control flies (30 days, Smurf + fly) due to a loss of intestinal integrity (*Figure 3—figure supplement 1A and B*). Although we found no significant intestinal barrier dysfunction in young *Atg9* mutants, there was a dramatic increase of Smurf + flies in aged *Atg9* mutant animals, compared to controls (*Figure 3—figure supplement 1B*).

The *Drosophila* adult midgut epithelium homeostasis is maintained by self-renewing intestinal stem cells (ISC) that divide asymmetrically to generate renewed ISCs and enteroblasts (EB) (*Micchelli and Perrimon, 2006*; *Ohlstein and Spradling, 2006*). The EB further differentiate into either absorptive enterocytes (EC) or secretory enteroendocrine cells (EE). In wild-type or *Atg9* heterozygous mutant flies, the intestinal epithelium consistently showed a tight, polarized monolayer (*Figure 3C,D*). In *Atg9* mutant flies, the epithelium was also a monolayer, but composed of dramatically enlarged cells and abnormal apical membrane protrusions that often expanded into the midgut lumen (*Figure 3C–E*). Notably, the aberrant midgut defects in *Atg9* mutants can be fully rescued by the *Atg9* genomic transgene (*Figure 3C–E*), further demonstrating that the intestinal defects were a direct consequence of the *Atg9* mutation.

To investigate whether the aberrant posterior midgut enlargement in *Atg9* mutants was due to an increase in cell growth and proliferation, we measured the ISC mitotic index in the *Atg9* mutant adult midgut. Immunostaining with an antibody for the mitotic marker phospho-histone 3 (PH3) in the whole midgut revealed no statistical difference in the number of PH3-positive cells between *Atg9* null and control flies at both 5 days and 30 days of age (*Figure 3F*). Moreover, the *Atg9* mutant midgut showed a similar number of total intestinal cells, Delta-positive (ISC-specific marker) cells and Pros-positive (EE marker) cells, compared with controls (*Figure 3G*). Together, these results indicate that Atg9 is not required for the regulation of ISC proliferation under normal conditions.

## Atg9 acts in ECs to regulate cell growth

To gain insights on the cell type requirement of Atg9 function, we expressed *Atg9^RNAi* with *Dl-Gal4* driver in ISCs, *Su(H)GBE-Gal4* in EBs, and *Myo1A-Gal4* (*NP1-Gal4*) in ECs. While *Atg9* depletion in ISCs and EBs did not cause any observable defects in the midgut, *Atg9* depletion in ECs resulted in pronounced defects in the midgut epithelium, with a markedly increased cell size and aberrant cell morphology (*Figure 4A*). We further utilized the TARGET system (*McGuire et al., 2004*) to specifically knockdown *Atg9* in adulthood. The negative control was flies maintained at 18°C, and the positive control was flies shifted to 29°C within 24 hr after eclosion to inactivate Gal80^ts and enable expression of the RNAi targeting *Atg9* gene. Because the midgut defects was only observed when *Atg9^RNAi* was induced by temperature shift in adult flies with *Tub-Gal4; Tub-Gal80^ts* as driver, the aberrant midgut enlargement is likely not caused by defects during development (*Figure 4B*). Moreover, although the visceral muscle defects were observed in *Atg9* mutants (*Figure 3B*), we found that, like *Atg9* depletion in ISCs and EBs, ablation of *Atg9* with muscle-specific *How-Gal4^ts* did not cause any observable defects in the adult midgut (*Figure 4—figure supplement 1*). To further explore the role of Atg9 in adult midguts, we generated *Atg9* mitotic clones using the heat shock-inducible Flp-FRT system. Mosaic analysis of *Atg9* mutant clones revealed a dramatic increase in cell size of *Atg9* mutant cells (*Figure 4C*, GFP-negative, encircled). Next, we utilized the MARCM (mosaic analysis with a repressible cell marker) technique to generate clones of cells homozygous for the *Atg9* mutation (*Figure 4D*, marked by GFP). Immunostaining with Pdm1 (an EC specific marker) (*Lee et al., 2009*) revealed that many of the enlarged GFP-positive cells in *Atg9* mutant clone were stained positive for Pdm1 (*Figure 4D*), suggesting that Atg9 acts largely in ECs to regulate cell growth.

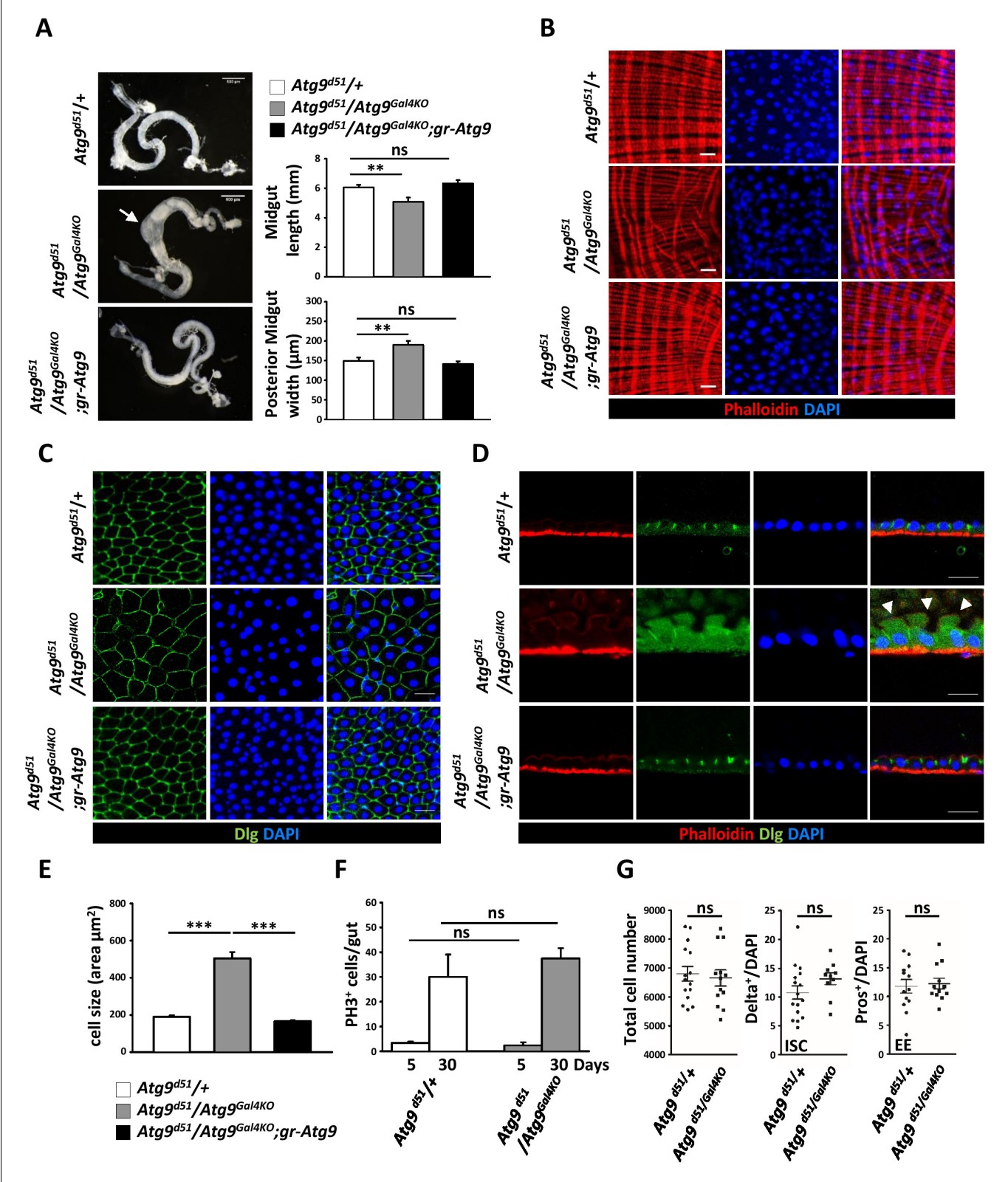

**Figure 3.** Atg9 is required for adult *Drosophila* midgut morphogenesis. (**A**) *Atg9* mutant midguts are shortened and display enlargement in the posterior region (arrow). Scale bar: 500 μm. Quantification of adult midgut length and posterior midgut width of control and *Atg9* mutant flies. n = 10, **p<0.01. (**B**) Phalloidin staining of midgut visceral muscles revealed that loss of *Atg9* leads to disruption of actin filaments. Scale bar: 20 μm. (**C–D**) Optical sections of control and the *Atg9* mutant midgut epithelium layer stained with anti-Dlg showing that *Atg9* mutants display abnormally enlarged

*Figure 3 continued on next page*

Figure 3 continued

cells with apical protrusions (arrowheads) into the lumen. Scale bar: 20 µm. (E) Quantification of cell size (shown in panel C) in control and *Atg9* mutant posterior midgut. n ≥ 25, *p<0.05, **p<0.01. (F) Quantification of phospho-Histone3 positive (PH3[+]) cells per midgut of control and *Atg9* mutant flies at 5 days and 30 days of age. n ≥ 8. (G) Quantification of total midgut cell numbers, posterior midgut ISC (Delta[+]) and EE (Pros[+]) cell numbers of 5-day-old control and *Atg9* mutant adults. n ≥ 10. Data are mean ±s.e.m. ns, not statistically significant.
DOI: https://doi.org/10.7554/eLife.29338.008

The following source data and figure supplements are available for figure 3:

**Source data 1.** Quantification of midgut length and width, cell size, total cell number, and PH3+, Delta+, Pros+ cells per gut.
DOI: https://doi.org/10.7554/eLife.29338.010
**Figure supplement 1.** Loss of intestinal integrity in aged *Atg9* mutant flies.
DOI: https://doi.org/10.7554/eLife.29338.009
**Figure supplement 1—source data 1.** Percentage of Smurfs.
DOI: https://doi.org/10.7554/eLife.29338.011

To investigate the involvement of other *Drosophila Atg* genes in midgut cell growth and homeostasis, we systematically depleted *Atg1*, *Atg13*, *Atg17/FIP200*, *Atg9*, *Atg7*, *Atg12*, *Atg16*, *Atg18*, and *Vps34* with *NP1-Gal4; tubulin-Gal80^ts* as driver. Surprisingly, only RNAi targeting *Atg9* and components of the Atg1 kinase complex, but not other *Atg* genes, caused prominent defects in the midgut epithelium (*Figure 5A,B*). Similar to the *Atg9* mutant, depletion of *Atg1*, *Atg13*, and *Atg17* showed enlarged and disorganized midgut epithelial cells. Moreover, we found that knockdown of *Atg1*, *Atg13*, and *Atg17* in the adult fly also caused intestinal barrier dysfunction and shortened lifespan (*Figure 5—figure supplement 1A and B*). We further confirmed that autophagy activity is efficiently blocked by temporal knockdown of these *Atg* genes (*Figure 5—figure supplement 2A and B*). Studies in yeast have shown that Atg9 is a direct target of Atg1 kinase during early autophagosome formation (*Papinski et al., 2014*). Activation of the Atg1 complex recruits and tethers Atg9-containing vesicles for the initiation of autophagy (*Rao et al., 2016*). In *Drosophila*, we have previously shown that ablation of *Atg9* suppresses Atg1-induced eye roughness and wing vein defects (*Tang et al., 2013*). We thus investigated whether Atg1 also genetically interacts with Atg9 in the adult midgut. Indeed, we found that overexpression of *Atg1* can rescue the midgut defects caused by depletion of *Atg9*, whereas *Atg1* depletion enhanced the phenotype (*Figure 5C*). These results highlight the critical role of Atg9 and the Atg1 kinase complex in maintaining adult midgut epithelium homeostasis.

## Functional interaction between Atg9 and the TOR pathway

The target of rapamycin (TOR) signaling pathway has been shown to regulate cell growth and proliferation (*Miron and Sonenberg, 2001*; *Zhang et al., 2000*). TOR is activated by the phosphatidylinositol 3-kinase (PI3K)/AKT pathway in response to nutrients or growth factors such as insulin stimulation (*Dibble and Cantley, 2015*; *Oldham and Hafen, 2003*). The activated TOR kinase phosphorylates ribosomal protein S6 kinase (S6K) and the eukaryotic translation initiation factor 4E-binding protein (4EBP) to regulate protein translation and cell size (*Katewa and Kapahi, 2011*). Studies in *Drosophila* and mammals have shown that loss of *Atg1* and *Atg17* can provoke TOR/S6K-dependent cell growth and development (*Kim et al., 2013*; *Lee et al., 2007*; *Scott et al., 2007*). This prompted us to investigate whether TOR signaling is activated in *Atg9* mutants. Intriguingly, Western blot analysis showed a marked increase in S6K and 4EBP phosphorylation levels in the midguts of *Atg9* mutants, compared to controls (*Figure 6A*). Inhibition of TOR activity by feeding *Atg9* mutant flies with rapamycin effectively suppressed the enlarged midgut cell size and aberrant epithelial morphology of the *Atg9* mutants (*Figure 6B and C*). Moreover, we found that rapamycin treatment significantly rescued the intestinal barrier dysfunction (*Figure 6—figure supplement 1A*), but not the lifespan defects of *Atg9* mutants (*Figure 6—figure supplement 1B*). We next checked whether modulation of the components of TOR signaling could rescue the Atg9 midgut defects. Overexpression of the TOR negative regulator *TSC1-TSC2* (the tuberous sclerosis complex 1 and 2), the dominant-negative TOR (*TOR^TED*), the dominant-negative S6K (*S6K^KQ*), or knockdown of TOR activator Rheb (*Rheb^RNAi*) strongly suppressed *Atg9^RNAi*-induced midgut defects (*Figure 6D–H,L*). Whereas overexpression of TOR activator Rheb or depletion of *TSC1* or *TSC2* enhanced *Atg9^RNAi*-induced midgut defects (*Figure 6I–K,L*). Moreover, ablation of components of the insulin receptor

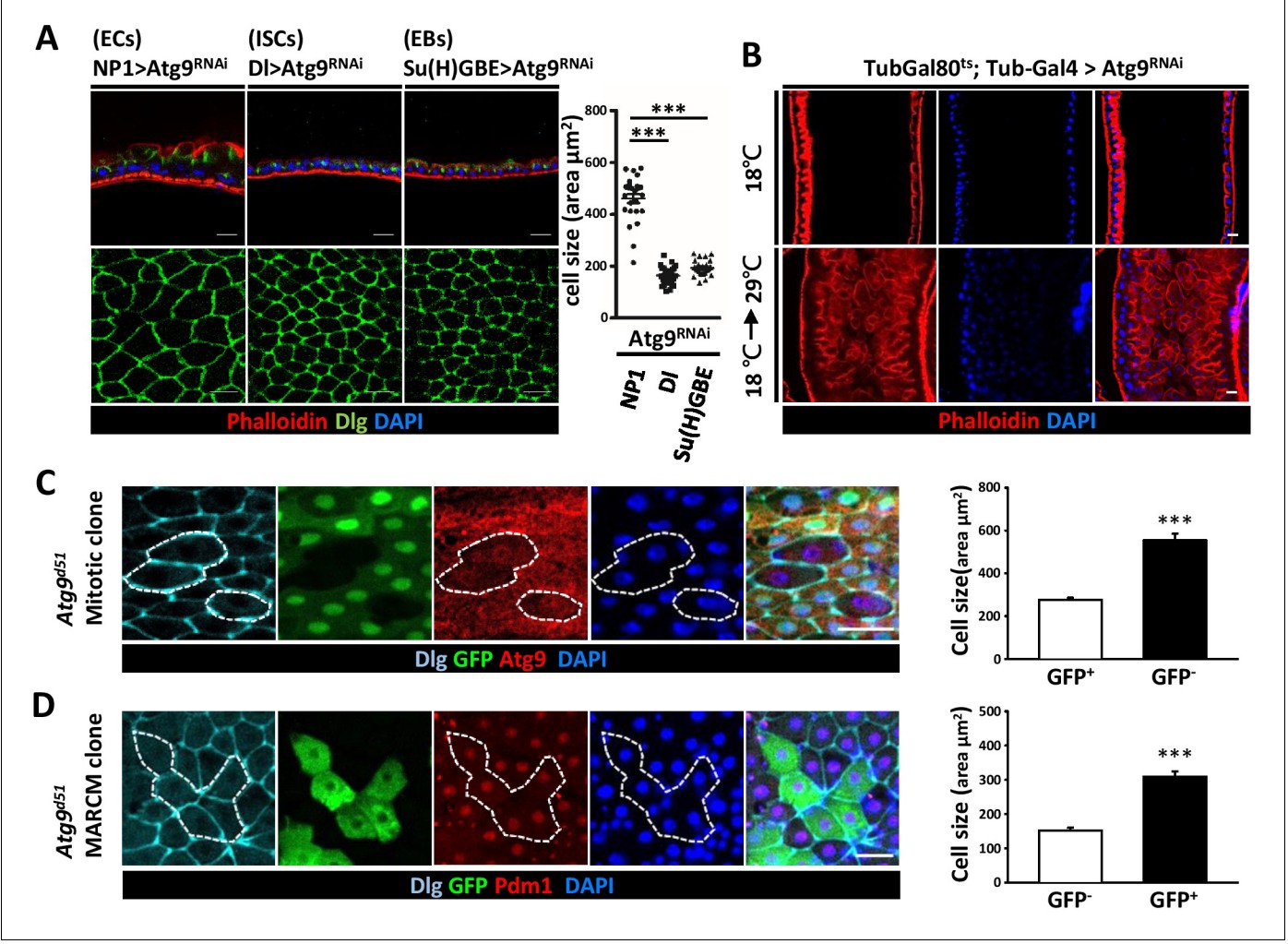

**Figure 4.** Loss of *Atg9* leads to enlarged enterocytes. (A) Expression *Atg9^RNAi* in ISCs, EBs and ECs with *Dl-Gal4, Su(H)GBE-Gal4*, or *NP1-Gal4*, respectively. Ablation of *Atg9* in ECs, but not ISCs or EBs, caused enlarged cell size. n ≥ 25, ***p<0.001. (B) Temporal control of *Atg9^RNAi* expression using the *Gal80^ts; Tub-Gal4* inducible system. The flies were either maintained at 18°C throughout development or shifted to 29°C after eclosion for 5 days to inactivate *Gal80^ts* and enable expression of the RNAi targeting *Atg9*. (C) Clonal analysis in adult midgut using Flp-FRT-mediated recombination revealed that *Atg9^d51* mutant cells (marked by lack of GFP and Atg9 expression) are larger than the controls (GFP-positive cells). n ≥ 17, ***p<0.001. (D) MARCM analysis showed that the enlarged *Atg9^d51* mutant cells (marked by GFP) are Pdm1 positive EC cells. n ≥ 21, ***p<0.001. Scale bar: 20 μm. Genotypes: (C) *hsFLP; FRT42D Ubi-GFP/FRT42D Atg9^d51* (D) *hsFLP; FRT42D tubGal80/FRT42D Atg9^d51; Tub-Gal4/UAS-mCD8GFP*.

DOI: https://doi.org/10.7554/eLife.29338.012

The following source data and figure supplements are available for figure 4:

**Source data 1.** Quantification of cell size.
DOI: https://doi.org/10.7554/eLife.29338.015
**Figure supplement 1.** *Atg9* depletion in visceral muscle does not cause any observable defects in the midgut.
DOI: https://doi.org/10.7554/eLife.29338.013
**Figure supplement 2.** *Atg9* depletion does not affect cell size of larval imaginal discs.
DOI: https://doi.org/10.7554/eLife.29338.014

(InR)-PI3K-AKT pathway, which is upstream of TOR, also rescues *Atg9^RNAi*-induced midgut defects (*Figure 6—figure supplement 2*). Taken together, these results suggest that Atg9 antagonizes TOR signaling in regulating cell growth and tissue homeostasis in the adult midgut.

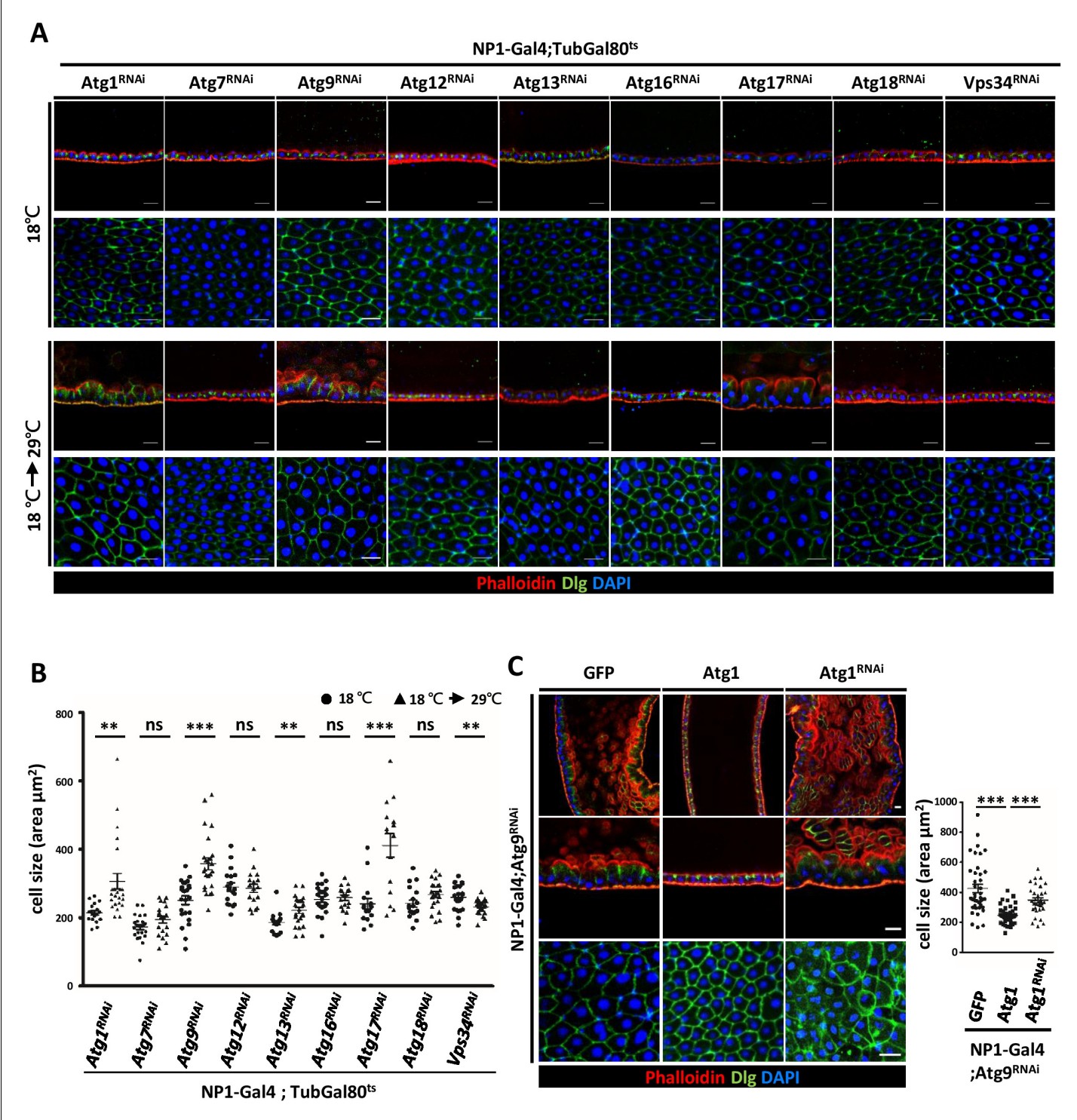

**Figure 5.** Components of Atg1 kinase complex are required for adult midgut epithelium homeostasis. (**A**) Systematic knock-down of *Drosophila Atg1*, *Atg7*, *Atg9*, *Atg12*, *Atg13*, *Atg16*, *Atg17*, *Atg18* and *Vps34* in adult midgut with the EC-specific driver *NP1-Gal4; Gal80$^{ts}$*. The flies were either kept at 18°C throughout development or shifted to 29°C after eclosion for 5 days to inactivate *Gal80$^{ts}$* and enable expression of the RNAi targeting *Atg* genes. Ablation of *Atg1*, *Atg13*, and *Atg17*, but not other *Atg* genes, caused increased cell size. (**B**) Quantification of posterior midgut cell size (shown in A) of *Atg* depleted flies. n ≥ 15, data are mean ±s.e.m. **p<0.01, ***p<0.001. ns, not statistically significant. (**C**) Overexpression of *Atg1* suppressed *Atg9$^{RNAi}$*-induced adult midgut defects. n ≥ 30, ***p<0.001. Scale bar: 20 μm.

DOI: https://doi.org/10.7554/eLife.29338.016

*Figure 5 continued*

The following source data and figure supplements are available for figure 5:

**Source data 1.** Quantification of cell size.
DOI: https://doi.org/10.7554/eLife.29338.019
**Figure supplement 1.** Depletion of *Atg1, Atg13,* or *Atg17* in adult fly causes intestinal barrier dysfunction and shortened lifespan.
DOI: https://doi.org/10.7554/eLife.29338.017
**Figure supplement 1—source data 1.** Survival rate and percentage of Smurfs.
DOI: https://doi.org/10.7554/eLife.29338.020
**Figure supplement 2.** Temporal knockdown of *Atg* genes impairs autophagy.
DOI: https://doi.org/10.7554/eLife.29338.018
**Figure supplement 2—source data 1.** Quantification of Atg8 dots.
DOI: https://doi.org/10.7554/eLife.29338.021

## Atg9 interacts with Patj and TSC2 to regulate midgut cell growth

To understand the mechanism by which Atg9 negatively regulates TOR signaling and cell growth, we performed a pull down assay using GST-fused C-terminus (residues 668–845) of Atg9 and liquid chromatography-tandem mass spectrometry (LC-MS/MS) to identify Atg9-interacting proteins. Among the proteins that were identified to interact with Atg9, we focused on Pals1-associated tight junction protein (Patj). While the multi-PDZ domain containing protein Patj forms a complex with the apical polarity protein Crumbs (Crb) and Stardust (Sdt; Pals1), several studies have indicated that Patj is not essential for apical basal polarity in *Drosophila* (*Pénalva and Mirouse, 2012*; *Sen et al., 2012*; *Zhou and Hong, 2012*). In mammals, Patj binds to tight-junction associated proteins such as Pals1, Claudin 1, and ZO-3, and regulates tight junction formation and cell migration (*Roh et al., 2002*; *Shin et al., 2007*). Interestingly, recent findings have shown that Patj can interact directly with TSC2 and depletion of Patj leads to increased TOR activity in human intestinal epithelial cells (*Massey-Harroche et al., 2007*; *Rosner et al., 2008*), suggesting that Patj may regulate TOR signaling through its interaction with TSC2. We thus performed GST pull-down and co-immunoprecipitation assays to confirm the interaction between Atg9 and Patj. As shown in *Figure 7A*, GST-Atg9-C, but not GST, efficiently interacted with Patj. To determine whether full-length Atg9 interacts with Patj, HEK293 cells were transiently transfected with Flag-tagged Atg9 (Flag-Atg9) and Myc-tagged Patj (Myc-Patj). Immunoblotting of the anti-Flag immunoprecipitates from cell lysates showed that Patj co-precipitated with Atg9 (*Figure 7B*). Similarly, a reciprocal co-immunoprecipitation experiment with anti-Myc antibody revealed an interaction between Atg9 and Patj (*Figure 7C*). Moreover, co-immunoprecipitation assays showed that Patj can interact with TSC2 in *Drosophila* S2 cells (*Figure 7D*). We next determined whether Patj could genetically interact with Atg9 in maintaining intestinal homeostasis. Indeed, similar to the *Atg9* mutant, we found that depletion of *Patj* with *NP1-Gal4* results in aberrant intestinal epithelium (*Figure 7E*). Overexpression of *Patj* in the midgut largely rescued the *Atg9* depletion-induced midgut defects (60%, n = 24) but not the midgut defects caused by *Atg1* or *Atg17* depletion (*Figure 7F* and *Figure 7—figure supplement 1A and B*), whereas *Patj* depletion exacerbated the midgut defects caused by *Atg9* ablation (*Figure 7F*). We have previously shown that ablation of *Atg9* in EC cells leads to increased lethality in response to paraquat ingestion (*Tang et al., 2013*). Strikingly, ectopic expression of *Patj* in EC cells significantly rescued the paraquat-induced lethality of *Atg9* knockdown animals (*Figure 7—figure supplement 1C*).

Several studies have shown that TSC2 is a short-lived protein and is readily targeted for degradation (*Chong-Kopera et al., 2006*; *Hu et al., 2008*). We thus investigated whether Atg9 could interact with and regulate TSC2 stability. As shown in *Figure 8A*, Atg9 specifically interacted with TSC2 by immunoprecipitation experiments, and depletion of *Atg1* does not affect the association between Atg9, Patj and TSC2 (*Figure 8—figure supplement 1A*). More strikingly, clonal depletion of *Atg9* (RFP-positive cells) in the adult midgut caused a marked decrease in TSC2 levels (*Figure 8B*), whereas TSC2 levels were not affected in *Atg1* or *Atg17* knockdown cells (*Figure 8—figure supplement 1B*). Immunoblotting analysis showed that TSC2 level is markedly decreased in the midgut of *Atg9* mutants, compared with controls (*Figure 8C*). Collectively, our results suggest that Atg9 antagonizes TOR signaling by interacting with Patj-TSC2 and regulates TSC2 stability (*Figure 8D*).

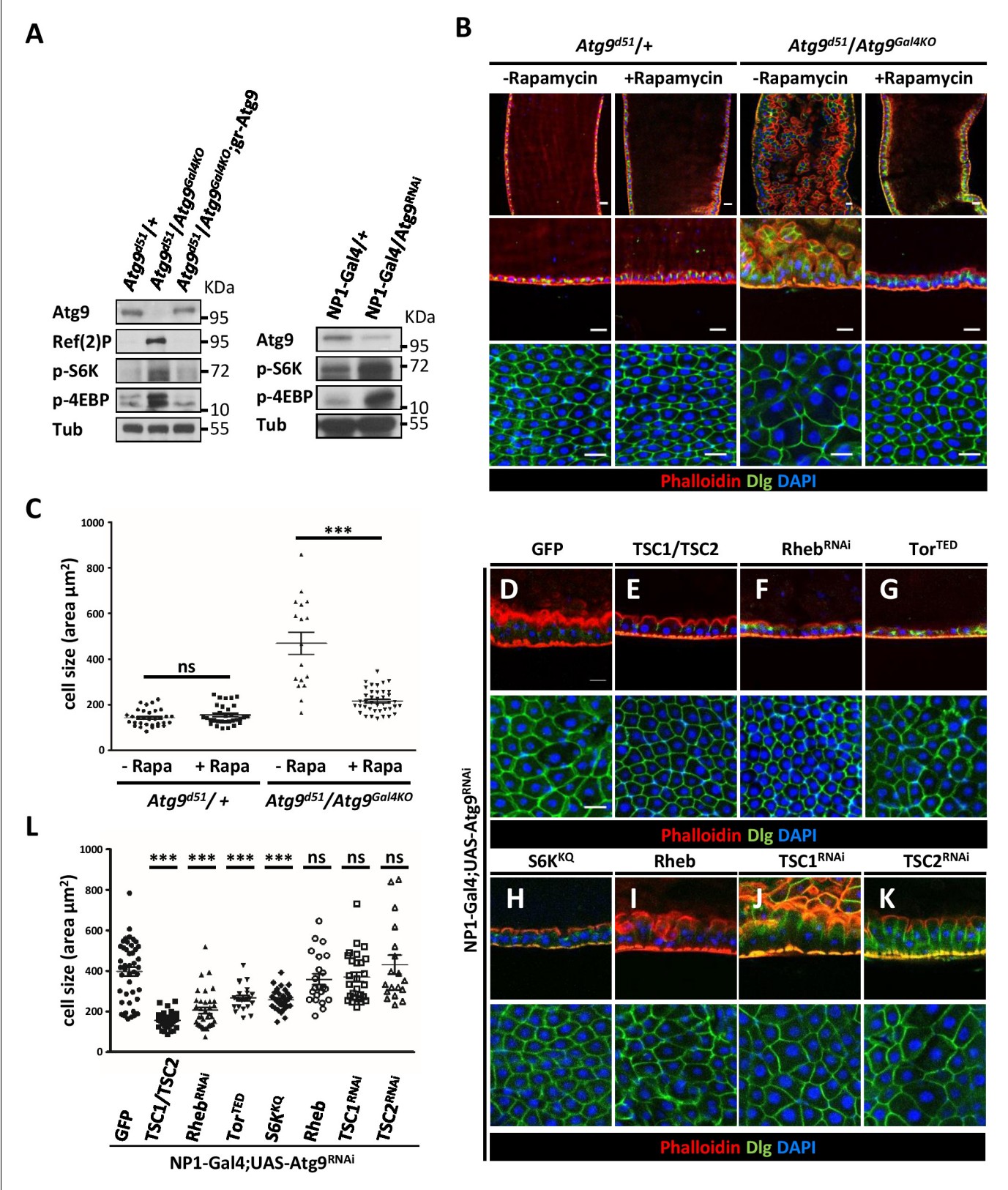

**Figure 6.** Loss of *Atg9* enhances TOR activity in *Drosophila* adult midgut. (**A**) The adult midguts of denoted genotypes were dissected, lysed, and subjected to Western blot analysis with anti-Atg9, anti-Ref(2)P, anti-p-S6K, anti-p-4EBP and anti-tubulin antibodies. (**B**) Inhibition of TOR activity by feeding flies with rapamycin rescued *Atg9* mutant midgut defects. (**C**) Quantification of posterior midgut cell size shown in (**B**). n ≥ 17, data are

*Figure 6 continued on next page*

*Figure 6 continued*

mean ±s.e.m. \*\*\*p<0.001. (D–K) Atg9 genetically interacts with components of the TOR signaling pathway. The *Atg9*[RNAi]-induced midgut defects (D) could be suppressed by the coexpression of *TSC1-TSC2* (E), *Rheb*[RNAi] (F), dominant-negative TOR (*TOR*[TED]) (G), or dominant-negative S6K (*S6K*[KO]) (H), whereas coexpression of TOR activator *Rheb* (I) or knock-down of *TSC1* (J) or *TSC2* (K) could not rescue the *Atg9*[RNAi]-induced midgut defects. Genetic analyses were performed for three times with 100% penetrance of the phenotype. (L) Quantification of posterior midgut cell size shown in (D–K). n ≥ 18, data are mean ±s.e.m. \*\*\*p<0.001. ns, not statistically significant. Scale bar: 20 μm.

DOI: https://doi.org/10.7554/eLife.29338.022

The following source data and figure supplements are available for figure 6:

**Source data 1.** Quantification of cell size.

DOI: https://doi.org/10.7554/eLife.29338.025

**Figure supplement 1.** Rapamycin treatment rescues the intestinal barrier dysfunction of *Atg9* mutants.

DOI: https://doi.org/10.7554/eLife.29338.023

**Figure supplement 1—source data 1.** Survival rate and percentage of Smurfs.

DOI: https://doi.org/10.7554/eLife.29338.026

**Figure supplement 2.** Atg9 genetically interacts with components of the insulin receptor/phosphoinositide 3-kinase (InR/PI3K) signaling pathway.

DOI: https://doi.org/10.7554/eLife.29338.024

**Figure supplement 2—source data 1.** Quantification of cell size.

DOI: https://doi.org/10.7554/eLife.29338.027

## Discussion

In this study, we generated *Drosophila Atg9* null mutants to determine the developmental and physiological function of Atg9. Similar to other autophagy mutants such as *Atg7* and *Atg17/Fip200* null flies (*Juhász et al., 2007*; *Kim et al., 2013*), *Atg9* mutants exhibit severe defects in *autophagy*, shortened lifespan, impaired motility, and hypersensitivity to stresses. *Atg9* loss-of-function also leads to aberrant midgut enlargement and intestinal barrier dysfunction. Interestingly, we found that, unlike *Atg9* mutant, depletion of *Atg7* in adult flies did not cause aberrant midgut enlargement or intestinal barrier dysfunction (*Figure 5* and *Figure 5—figure supplement 1A*). It is possible that the adult midgut defects may contribute to the much shorter lifespan of *Atg9* mutants. Our findings indicate that Atg9 not only acts as a key regulator in autophagy but also functions in maintaining adult *Drosophila* midgut homeostasis.

The *Drosophila* adult midgut is composed of a monolayer of epithelial cells including nutrient absorbing enterocytes (ECs), secretory enteroendocrine (EE) cells, and multipotent intestinal stem cells (ISCs) (*Lemaitre and Miguel-Aliaga, 2013*). A number of conserved signaling pathways, including insulin, Notch, EGFR, Wingless (Wg)/Wnt, Hippo, TOR, and JAK-STAT pathways, have been shown to be involved in the regulation of ISC proliferation and in the maintenance of tissue homeostasis of the *Drosophila* midgut (*Guo et al., 2016*; *Jiang et al., 2016*). The enlarged adult midguts observed in *Atg9* mutants may be due to dysregulation of cell proliferation and cell growth. ISCs are the only dividing cells in the *Drosophila* adult midgut and play an essential role in maintaining tissue homeostasis. However, we found that the total number of intestinal cells and Delta[+] ISC population were not affected in *Atg9* mutants. PH3 staining of control or *Atg9* mutant midguts at young (5 day) or old (30 day) stages showed similar numbers of mitotic ISCs in both animals, thus indicating that loss of *Atg9* does not affect ISC proliferation.

Insulin and TOR signaling are conserved nutrient-sensing pathways involved in regulating cell growth, metabolism and tissue homeostasis (*Oldham and Hafen, 2003*). Recent studies have reported that these pathways regulate enterocyte growth and endoreduplication in the adult *Drosophila* midgut (*Amcheslavsky et al., 2011*; *Kapuria et al., 2012*; *Xiang et al., 2017*). As the loss of *Atg9* leads to a marked increase in TOR activity, while inhibition of insulin/TOR signaling rescues *Atg9* midgut defects, Atg9 may act as a negative regulator of TOR-mediated cell growth. It is also interesting to note that although *Atg9* depletion caused a dramatic enlargement in adult midgut ECs, we did not find an increase in larval disc cell size (*Figure 4—figure supplement 2*). It has been shown that the normal development and tissue homeostasis is maintained by a delicate coordination between cell growth and division (*Jorgensen and Tyers, 2004*; *Zeng et al., 2013*). One possible explanation for this difference is that ECs in adult midgut are non-dividing differentiated cells, whereas larval disc cells are active dividing cells and therefore maintain a moderate cell size.

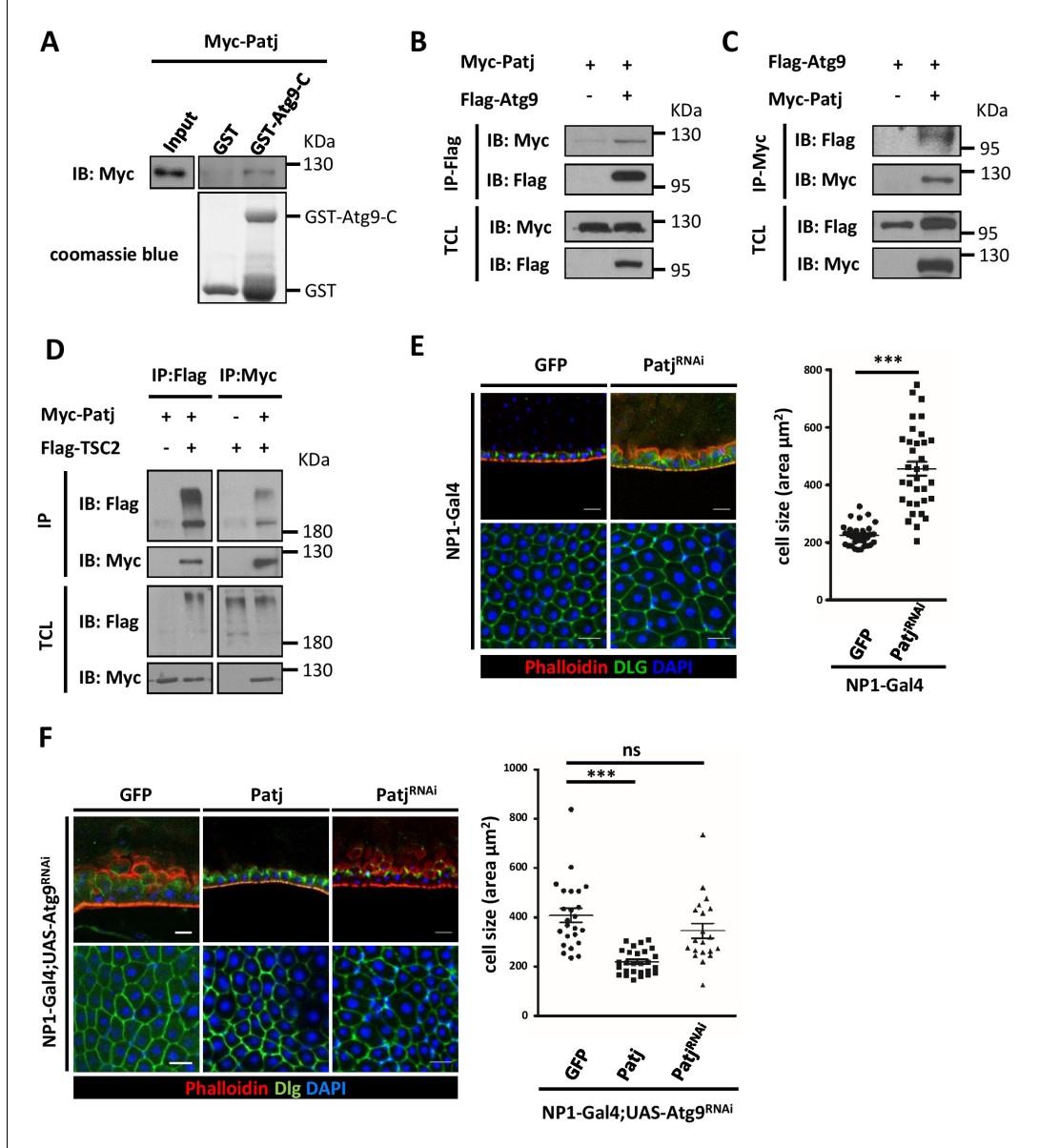

**Figure 7.** Atg9 interacts with Patj to regulate midgut cell growth. (**A**) Lysates of cells expressing Myc-Patj were incubated with GST or GST–Atg9-C (amino acids 668–845) in GST pull-down assays. The pull-down products and input Myc-Patj were analyzed by Western blots with the anti-Myc antibody. (**B–C**) HEK293T cells transfected with Flag-Atg9 and Myc-Patj were subjected to immunoprecipitation with anti-Flag (**B**) or anti-Myc (**C**) antibody. The immunoprecipitates and total cell lysates (TCL) were analyzed by Western blot with antibodies as indicated. (**D**) S2 cells transfected with pWA-Gal4, pUAS-Flag-TSC2 and pUAS-Myc-Patj were subjected to immunoprecipitation with anti-Flag or anti-Myc antibody. The immunoprecipitates and total cell lysates were analyzed by Western blot with antibodies as indicated. (**E**) Depletion of *Patj* with *NP1-Gal4* resulted in aberrant midgut epithelium and increased EC cell size. n ≥ 34, data are mean ±s.e.m. ***p<0.001. (**F**) Patj genetically interacts with Atg9. Overexpression of *Patj* suppressed the *Atg9* depletion-induced midgut defects (60% penetrance, n = 24). The cell size of posterior midgut ECs of each genotype was quantified. n ≥ 20, data are mean ±s.e.m. ***p<0.001. Scale bar: 20 μm.

DOI: https://doi.org/10.7554/eLife.29338.028

The following source data and figure supplements are available for figure 7:

**Source data 1.** Quantification of cell size.
DOI: https://doi.org/10.7554/eLife.29338.030

**Figure supplement 1.** Atg9 genetically interacts with Patj.
DOI: https://doi.org/10.7554/eLife.29338.029

**Figure supplement 1—source data 1.** Survival rate and quantification of cell size.
DOI: https://doi.org/10.7554/eLife.29338.031

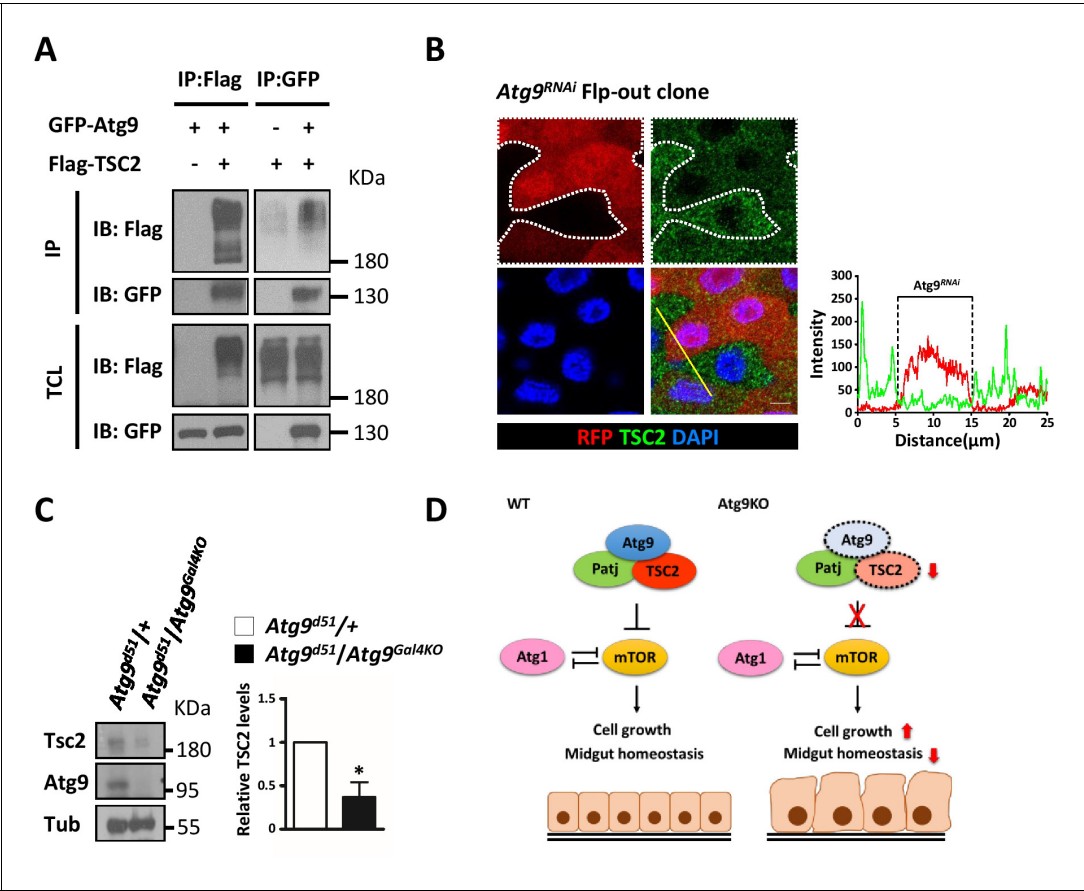

**Figure 8.** Atg9 interacts with TSC2 to regulate midgut cell growth. (**A**) S2 cells transfected with pWA-Gal4, pUAS-Flag-TSC2 and pUAS-GFP-Atg9 were subjected to immunoprecipitation with anti-Flag or anti-GFP antibody. The immunoprecipitates and total cell lysates were analyzed by Western blot with antibodies as indicated. (**B**) Clonal expression of *Atg9^RNAi* (marked in Red) in the adult midgut resulted in marked decrease in TSC2 levels (green). Line scan across the *Atg9^RNAi* clone to show the relative fluorescent intensities of TSC2 in control (dsRed-negative) and *Atg9* depletion (dsRed-positive) cells. Scale bar: 5 μm. (**C**) The adult midguts of denoted genotypes were dissected, lysed, and subjected to Western blot analysis with anti-Atg9, anti-TSC2 and anti-tubulin antibodies. N = 3, data are mean ±s.e.m. *p<0.05. (**D**) Model for the antagonistic effect of Atg9 on TOR signaling in the regulation of intestinal cell growth and midgut homeostasis in *Drosophila*. Genotypes: (**G**) *hsflp; UAS-Atg9^RNAi/+; Act-CD2-Gal4-UAS-dsRed..*
DOI: https://doi.org/10.7554/eLife.29338.032

The following source data and figure supplements are available for figure 8:

**Source data 1.** Quantification of fluorescent intensity and Western blots.
DOI: https://doi.org/10.7554/eLife.29338.034

**Figure supplement 1.** Atg9 interacts with Patj and TSC2 independent of Atg1.
DOI: https://doi.org/10.7554/eLife.29338.033

**Figure supplement 1—source data 1.** Quantification of fluorescent intensity.
DOI: https://doi.org/10.7554/eLife.29338.035

Besides Atg9, we found that depletion of components of the Atg1 kinase complex also resulted in enlarged adult midgut ECs. While TOR inhibits Atg1/Ulk1-mediated autophagy induction under nutrient-rich conditions, recent reports have shown that Atg1 and Ulk1 negatively regulate TOR signaling in *Drosophila* and mammalian cells, respectively (*Lee et al., 2007*; *Scott et al., 2007*). Although Atg9 was found to be a downstream target of Atg1 during autophagy, several lines of evidence suggest that Atg1 inhibits TOR independent of Atg9. First, our genetic results showed that overexpression of *Atg1* rescued *Atg9^RNAi*-induced midgut defects; in contrast, overexpression of *Atg9* or *Patj* could not suppress the midgut defects caused by *Atg1* depletion (*Figure 7—figure supplement 1A*). Second, co-immunoprecipitation assays showed that Atg9 can still interact with Patj and TSC2 in *Atg1* knockdown S2 cells (*Figure 8—figure supplement 1A*). Thirdly, clonal analysis showed that although the TSC2 level was markedly decreased in *Atg9* knockdown cells, TSC2

levels were not affected in *Atg1* or *Atg17* knockdown cells, compared with controls (*Figure 8—figure supplement 1B*). In mammals, it has been shown that Atg1/Ulk1 inhibits TOR signaling by phosphorylating Raptor and impairs substrate binding to Raptor (*Dunlop et al., 2011*; *Jung et al., 2011*), and the inhibitory effect of Ulk1 on TOR signaling occurred independently of TSC2 (*Jung et al., 2011*). Our results and findings from mammalian cells together strongly indicate that Atg9 inhibits TOR activity independent of Atg1.

The formation of the midgut epithelium in *Drosophila* depends on the establishment of cell polarity and adhesion (*Müller, 2000*). One striking phenotype observed in *Atg9* mutant midgut epithelium is the appearance of enlarged epithelial cells with aberrant apical membrane expansion. The *Drosophila* epithelial polarity is regulated by evolutionarily conserved polarity protein complexes including the Crumbs (Crb)/Stardust (Sdt)/Patj complex, the Bazooka (Baz)/Par6/aPKC complex and the Scribble/Discs large (Dlg)/Lethal giant larva (Lgl) complex (*Tepass, 2012*). While Crumbs and Par complexes are localized apically in epithelial cells for apical domain maintenance, the Scribble complex is localized in the basolateral region for the maintenance of the basolateral membrane. It has been reported that the Par proteins Par3/Baz, Par6, and aPKC are involved in regulating asymmetric cell division and the differentiation of adult *Drosophila* ISCs (*Goulas et al., 2012*). Here, we identified Patj as a novel interactor of Atg9. The scaffolding protein Patj interacts with both Crb-Std and Baz-Std apical polarity protein complexes, and has been shown to play a non-essential role in apical-basal polarity during *Drosophila* development (*Pénalva and Mirouse, 2012*; *Sen et al., 2015*; *Zhou and Hong, 2012*). However, the role of Patj in midgut epithelium formation and the molecular mechanisms underlying Atg9-Patj-TSC2-mediated intestinal cell growth remain unknown. Interestingly, like Atg9, Patj has been shown to cycle between plasma membrane and endosomal compartments in mammalian cells (*Heller et al., 2010*; *Wells et al., 2006*). Ablation of *Patj* results in accumulation of Crumbs3 in EEA-1 positive early endosomes (*Michel et al., 2005*). It will be exciting to determine how Atg9 coordinates with Patj to regulate TSC2 stability and intestinal epithelial homeostasis. Further research should aid our understanding in the regulation of TOR signaling by Atg9 and its role in human diseases.

## Materials and methods

### *Drosophila* strains and genetics

Flies were raised at 25°C following standard procedures unless otherwise noted. The following *Drosophila* strains were used: $Atg7^{d14}$, $Atg7^{d77}$ (*Juhász et al., 2007*), UAS-Atg1 (*Scott et al., 2007*), Dl-Gal4, Su(H)GBE-Gal4, (*Zeng et al., 2010*), $How-Gal4^{ts}$ (gift from Bruno Lemaitre, EPFL), NP1-Gal4, UAS-TSC1, UAS-TSC2 (*Tang et al., 2013*), UAS-Rheb (RRID:BDSC_9688), UAS-S6K$^{KQ}$ (RRID:BDSC_6911), UAS-Tor$^{TED}$ (RRID:BDSC_7013), UAS-Atg1$^{RNAi}$ (RRID:BDSC_44034), UAS-Rheb$^{RNAi}$ (RRID:BDSC_33966), UAS-Atg7$^{RNAi}$ (RRID:BDSC_34369), UAS-Atg9$^{RNAi}$ (RRID:BDSC_34901), UAS-Atg12$^{RNAi}$ (RRID:BDSC_34675), and Df(2R)Exel7142 (RRID:BDSC_7886) flies were obtained from the Bloomington Stock Center; UAS-Patj$^{RNAi}$ (RRID:FlyBase_FBst0473750), UAS-Atg13$^{RNAi}$ (RRID:FlyBase_FBst0475239), UAS-Atg16$^{RNAi}$ (RRID:FlyBase_FBst0477819), UAS-Atg17$^{RNAi}$ (RRID:FlyBase_FBst0476692), UAS-Atg18$^{RNAi}$ (RRID:FlyBase_FBst0477193), UAS-Vps34$^{RNAi}$ (RRID:FlyBase_FBst0472170), UAS-TSC1$^{RNAi}$ (RRID:FlyBase_FBst0454493), and UAS-TSC2$^{RNAi}$ (RRID:FlyBase_FBst0470276) were obtained from the Vienna *Drosophila* Resource Center.

### Generation of *Atg9* mutants

The $Atg9^{Gal4KO}$ mutant was generated by knock-in replacement of the *Atg9* genomic region with the Gal4 cassette as described previously (*Chan et al., 2011*). In brief, we first generated two 500 bp homology arms flanking the 20 kb *Atg9*-containing genomic fragment for first-round recombineering. The following primers were used in a SOEing PCR reaction:

LA500-fwd: 5'-ACAAGTTTGTACAAAAAAGCAGGCT TTGGCAGGCACACGACATTT −3'
LA500-rev: 5'-CACGCAGGATCCCTTCAATCCAGAGCAACAGG-3'
RA500-fwd: 5'-TTGAAGGGATCCTGCGTGGAACCCATCTTTGG-3'
RA500-rev: 5'-ACCACTTTGTACAAGAAAGCTGGGT TTGCATTTTGTTTGCTAAGT −3'

The 1 kb PCR product was then cloned into P[acman]-KO to generate P[acman]-KO-Atg9-1kb, and the resultant plasmid was digested with BamHI and transformed into DY380 cells containing the

*Atg9* genomic DNA. To replace the *Atg9* open-reading frame with the Gal4-RFP-Kan cassette, the following primers were used in second-round recombineering:

Atg9-fwd:5'-TCTCTTAGGAGAGTCAGCTGTTTGCTGAGAAGGTTCAGCAGAATCAAACAA-CAAAGAATTTTCCAACTTATACTATACAGCGATATAAATAGTCAGAACGGTTGACCTTGACG TTGGGCG-3'

Atg9-rev:5'-GTTTAGCTTAGTTTCAGATTAGTTTAGCTACGCACTAGACGACGTCGTTCGTTCG TTTACACTTTAAAATTTAGGTTAATCACTAATAGCAGAATGGGTGGTCCTTAGCTCTACAGGTGG-3'

The PCR product was transformed into DY380 cells containing P[acman]-KO-Atg9-20kb. The transformants containing P[acman]-Atg9-Gal4 were PCR-verified followed by sequencing. The confirmed P[acman]-Atg9-Gal4 was then injected into the embryo following standard transgenesis protocol. The transgenic *Atg9*$^{Gal4}$ allele was then excised in vivo and targeted to the endogenous atg9 locus to generate *Atg9*$^{Gal4KO}$ flies.

The *Atg9*$^{d51}$ mutant was generated using CRISPR/cas9-based genome editing (*Kondo and Ueda, 2013*). In brief, gRNA sequence GAGGGATGGTGCTCCAGGAA[CGG] was cloned into pBFv-U6.3. Cassette attPX-3-frame Stop-floxed 3xP3-RFP flanked by two *Atg9* homology arms (500 bp upstream and downstream of CRISPR targeting site) was cloned into pCR2-TOPO. *Atg9*-targeting gRNA plasmid and donor template for repair were co-injected into embryos of nanos-Cas9 expressing flies. Progeny flies carrying the selection marker of 3xP3-RFP were further validated by genomic PCR and sequencing. The CRISPR-mediated mutagenesis was performed with the help of WellGenetics, Inc. To induce mitotic clones, *hsFLP; FRT42D Ubi-GFP/FRT42D Atg9*$^{d51}$ flies were heat shocked in a 37°C water bath for 1 hr twice a day. MARCM clones were generated by placing *hsFLP; FRT42D tubGal80/FRT42D Atg9*$^{d51}$*; Tub-Gal4/UAS-mCD8GFP* flies in a 37°C water bath for 1 hr twice a day. The flies were kept at 25°C for seven additional days before dissection.

## RT-PCR

Total RNA was isolated from adult female flies using TRIzol (Invitrogen). cDNA was synthesized from 1 ug RNA using the Transcriptor First Strand cDNA Synthesis Kit (Roche) according to the manufacturer's instructions. The following pairs of specific primers were used: 5'-TTGTCCAGATCCGAATCC TC-3' (Atg9-L); 5'-TCGTCTGGCTACTTGCCTTT-3' (Atg9-R); 5'-TTGTCTGGGCAAGAGGATCAG-3' (Actin5C-L); 5'-ACCACTCGCACTTGCACTTTC-3' (Actin5C-R).

## Plasmids and antibodies

The pUAS-GFP-Atg9 transgene was generated by PCR amplification Atg9 from RE14003 into the pUAST vector. The Myc-tagged Patj expression plasmid was generated by PCR amplification of Patj from LD22238, then cloned into the pUAST or pcDNA3.1 vector. *Drosophila* pUAS-Flag-TSC2 was kindly provided by Jun Hee Lee (University of Michigan). For the Atg9 genomic rescue construct, the Atg9 genomic locus flanked by 1 kb each of upstream and downstream genomic sequence was cloned into pCaSpeR4. Antibodies used for the study were: anti-Dlg (1:100, DSHB, University of Iowa), anti-Delta (1:100, RRID:AB_2056641), and anti-Prospero (1:100, RRID:AB_528440), anti-Atg9 (1:100) (*Tang et al., 2013*), anti-TSC2 (1:100, gift from Aurelio Teleman, German Cancer Research Center) (*Tsokanos et al., 2016*), anti-Pdm1 (1:100, gift from Xiaohang Yang, Zhejiang University), anti-Atg8 (1:100, RRID:AB_297935), anti-Ref(2)p (1:500, Abcam, Cat# ab178440), anti-Ub (1:100, RRID:AB_10691572), anti-phospho-S6K (1:1000, Cell Signaling, Cat# 9209), and anti-phospho-4EBP (1:1000, Cell Signaling, Cat# 2855), anti-GFP (1:100, RRID:AB_1563142), anti-Myc (1:500, RRID:AB_2298152), anti-PH3 (1:1000, RRID:AB_477043), anti-Flag (1:1000, RRID:AB_2687448), and anti-tubulin (1:5000, RRID:AB_1844090).

## Cell culture, transfection and immunoprecipitation

*Drosophila* S2 cells were cultured at 25°C in Schneider's *Drosophila* medium (Thermo Fisher) containing 10% fetal bovine serum (FBS) and antibiotics. HEK293T cells were cultured at 37°C in Dulbecco's modified Eagle's medium (DMEM) medium (Thermo Fisher) supplemented with 10% FBS and antibiotics. S2 cells were transfected with Lipofectamine 2000 (Invitrogen), whereas HEK293T were transfected with PolyJet (SignaGen) according to the manufacturer's protocol. The dsRNAs were generated using the T7 RiboMAX Express RNAi System (Promaga). For immunoprecipitations, cells transiently transfected with the indicated plasmids were scraped from dishes in lysis buffer (50 mM

Tris-HCl pH 7.4, 150 mM NaCl, 1% Triton, 10% glycerol, 1 mM EDTA, 10 mM NaF, 1 mM PMSF, and protease inhibitor cocktail (Roche)). Cell lysates were immunoprecipitated with Myc or Flag antibody at 4°C overnight and protein G-Sepharose beads (GE Healthcare) at 4°C for 1 hr. These beads were washed three times with the lysis buffer.

## Immunofluorescence

For *Drosophila* midgut immunohistochemistry, female adult midguts were dissected in PBS and immediately fixed in 4% paraformaldehyde for 40 min at room temperature. After fixation, the samples were permeabilized in PBST (PBS containing 0.5% Triton X-100) for 10 min, then blocked in PBST containing 5% Normal Goat Serum (NGS) for 1 hr. The samples were incubated with primary antibodies in PBST-NGS at 4°C for 16 hr. On the following day, samples were washed with PBST and incubated with fluorescent-labeled secondary antibodies at 4°C overnight. Nuclei were stained using DAPI (1 µg/ml). Images were acquired using the Zeiss LSM510 or Olympus FV3000RS confocal laser scanning microscope. For lysotracker staining, *Drosophila* larval fat bodies dissected under fed or starved conditions were incubated for 45 s in 100 µM LysoTracker Green DND-26 (Thermo) with DAPI in PBS, and immediately photographed live on an Olympus BX61 fluorescence microscope.

## Lifespan, climbing, and stress resistance analyses

The lifespan assays were performed as described previously (*Chen et al., 2012*). Female or male flies were housed in groups of 20 and the flies were transferred to fresh food every 2–3 days until all were dead. Climbing assays (*Martinez et al., 2007*) and stress resistance experiments were performed as preciously described (*Juhász et al., 2007*; *Tang et al., 2013*). At least three independent measurements were performed for each experiment.

## Rapamycin feeding

Rapamycin feeding was performed as previously described (*Amcheslavsky et al., 2011*), with flies being incubated on the food medium containing 50 µM rapamycin (Merck) for 7 days.

## Statistical analyses

Statistical analysis was performed by Student's t test. Log-rank test was used for lifespan statistical analysis. Differences were considered significant if p values were less than 0.05 (*), 0.01 (**), or 0.001 (***).

## Acknowledgements

We thank Chun-Hong Chen, Gábor Juhász, Elisabeth Knust, Bruno Lemaitre, Jun Hee Lee, Thomas Neufeld, Henry Sun, Aurelio Teleman, Xiaohang Yang, Xiankun Zeng, the Bloomington Stock Center, Vienna *Drosophila* RNAi Center, Developmental Studies Hybridoma Bank, and Fly Core Taiwan for reagents and fly stocks. We thank Horng-Dar Wang for technical advice and Chiou-Yang Tang and WellGenetics for *Drosophila* embryo injection. We are grateful to Chi-Kuang Yao for helpful comments on the manuscript and Cindy Lee for English editing. This work was supported in part by the Ministry of Science and Technology of Taiwan (MOST105-2311-B-001–062-MY3 to GCC and MOST104-2311-B-002–017-MY3 to CCC) and the Academia Sinica Career Development Award (101CDA-L04 to GCC).

## Additional information

### Funding

| Funder | Grant reference number | Author |
| --- | --- | --- |
| Ministry of Science and Technology, Taiwan | MOST104-2311-B-002-017-MY3 | Chih-Chiang Chan |
| Ministry of Science and Technology, Taiwan | MOST105-2311-B-001-062-MY3 | Guang-Chao Chen |
| Academia Sinica | 101CDA-Lo4 | Guang-Chao Chen |

The funders had no role in study design, data collection and interpretation, or the decision to submit the work for publication.

## Author contributions

Jung-Kun Wen, Conceptualization, Investigation, Formal analysis, Validation, Visualization, Software, Methodology, Data curation, Writing—review and editing; Yi-Ting Wang, Investigation, Formal analysis, Validation, Methodology, Visualization, Data curation; Chih-Chiang Chan, Conceptualization, Methodology, Resources, Writing—review and editing, Funding acquisition; Cheng-Wen Hsieh, Hsiao-Man Liao, Investigation, Formal analysis, Validation; Chin-Chun Hung, Methodology, Visualization, Software; Guang-Chao Chen, Conceptualization, Resources, Methodology, Formal analysis, Supervision, Data curation, Writing—original draft, Project administration, Funding acquisition, Writing—review and editing

## Author ORCIDs

Chih-Chiang Chan, https://orcid.org/0000-0003-2626-3805
Guang-Chao Chen, https://orcid.org/0000-0002-4980-4718

## Decision letter and Author response

Decision letter https://doi.org/10.7554/eLife.29338.038
Author response https://doi.org/10.7554/eLife.29338.039

# Additional files

## Supplementary files

• Transparent reporting form
DOI: https://doi.org/10.7554/eLife.29338.036

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
