## [Decision Letter]

Thank you for submitting your article "Atg9 antagonizes TOR signaling to regulate intestinal cell growth and epithelial homeostasis in *Drosophila*" for consideration by *eLife*. Your article has been favorably evaluated by K VijayRaghavan (Senior Editor) and three reviewers, one of whom is a member of our Board of Reviewing Editors.

The reviewers have discussed the reviews with one another and the Reviewing Editor has drafted this decision to help you prepare a revised submission.

Summary:

The authors report that Atg9 mutant flies show a reduction in lifespan, mobility, and susceptibility to starvation and oxidative stresses. Atg9 mutant flies also show enlarged midgut epithelial cells. Knockdown of Atg1 or Atg17, but not other core Atgs, causes a similar midgut phenotype. The TORC1 pathway is activated in Atg9 mutants because Atg9 interacts with PALS1-associated tight junction protein (Patj) and TSC2 to inactivate TORC1. Pharmacological and genetic inhibition of TOR and overexpression of Patj rescue the midgut phenotype in Atg9 mutants.

Overall, this study is novel and provides new insights into both autophagy and mTOR regulation. The data are mostly convincing and logically explained.

Essential revisions:

1) It is not completely clear whether the two autophagy factors, Atg9 and Atg1, regulate mTOR independently as shown in Figure 7. The following experiments would clarify this issue.

- Can the midgut phenotype of Atg1 or Atg17 knockdown flies be rescued by overexpression of Patj (Figure 7)?

- Is TSC2 unstabilized in Atg1 or Atg17 knockdown flies as in Atg9 knockdown flies (Figure 7)?

- Can the Atg9-Ptj-TSC2 form a complex normally in Atg1 knockdown flies?

2) Figure 5: Is autophagy efficiently blocked by temporal knockdown using the TARGET system. To strengthen the results, the authors should show that autophagic activity is indeed inhibited in these knockdown flies or examine adult midgut in pre-existing mutant flies such Atg7 mutants.

3) Figure 7: Depletion of Atg9 in the adult midgut causes a marked decrease in the TSC2 signal level in immunostaining, but it is not very quantitative. The authors should perform immunoblotting to compare the TSC2 protein level in control and Atg9 mutants, as performed in Figure 6.

4) It remains unclear whether the increased growth of EC in ATG9 mutants is really biologically important. When the increased EC size in ATG9 mutants is rescued (e.g. by TOR inhibition or Patj overexpression in EC cells), are the barrier function and the shortened lifespan restored? These will demonstrate that the increased EC size after a loss of ATG9 is functionally important in affecting the midgut epithelium and lifespan.

5) Some of the critical data should be quantified (e.g., Figure 1, Figure 1—figure supplement 1, Figure 5, and Figure 6) to strengthen this manuscript.

---

## [Author Response]

Essential revisions:1) It is not completely clear whether the two autophagy factors, Atg9 and Atg1, regulate mTOR independently as shown in Figure 7. The following experiments would clarify this issue.- Can the midgut phenotype of Atg1 or Atg17 knockdown flies be rescued by overexpression of Patj (Figure 7)?- Is TSC2 unstabilized in Atg1 or Atg17 knockdown flies as in Atg9 knockdown flies (Figure 7)?- Can the Atg9-Ptj-TSC2 form a complex normally in Atg1 knockdown flies?

We thank the reviewer for this comment. We have performed several new experiments to demonstrate that Atg9 and Atg1 regulate TOR activity independently. As shown in new Figure 7—figure supplement 1, we found that overexpression of Patj could not rescue the midgut defects caused by Atg1 or Atg17 depletion. Moreover, clonal analysis showed that although the TSC2 level was markedly decreased in Atg9 knockdown cells, TSC2 levels were not affected in Atg1 or Atg17 knockdown cells, compared with controls (new data, Figure 8—figure supplement 1). Finally, co-immunoprecipitation assays showed that Atg9 can still interact with Patj and TSC2 in Atg1 knockdown S2 cells (new data, Figure 8—figure supplement 1). In mammals, it has been shown that Atg1/Ulk1 inhibits TOR signaling by phosphorylating Raptor and impairs substrate binding to Raptor (Dunlop et al. 2011, Autophagy and Jung et al. 2011,Autophagy), and the inhibitory effect of Ulk1 on TOR signaling occurred independently of TSC2 (Jung et al. 2011, Autophagy). Our new results and findings from mammalian cells together strongly indicate that Atg9 inhibits TOR activity independent of Atg1.

2) Figure 5: Is autophagy efficiently blocked by temporal knockdown using the TARGET system. To strengthen the results, the authors should show that autophagic activity is indeed inhibited in these knockdown flies or examine adult midgut in pre-existing mutant flies such Atg7 mutants.

We thank the reviewer for this comment. To determine whether autophagy is efficiently blocked by temporal knockdown Atg genes using the TARGET system, we have knocked down Atg genes using *Tub-Gal4^ts^* and checked their effects on starvation-induced autophagy in the larval fat body. The negative control was flies maintained at 18°C, and the positive control was flies shifted to 29°C to inactivate *Gal80^ts^* and enable expression of the RNAi targeting Atg genes. As shown in new Figure 5—figure supplement 2, depletion of Atg1, Atg13, Atg17, Atg7, Atg12, Atg16, Atg18, and Vps34 impaired starvation-induced autophagy. We have previously shown that depletion of Atg9 and Atg12 blocked ROS-induced autophagy in adult midgut (Tang et al. 2013, Dev Cell). Similarly, we found that ROS-induced autophagy is efficiently blocked in the adult midgut of Atg7 mutants (Author response image 1).

**Author response image 1. respfig1:** Loss of Atg7 impairs paraquat-induced autophagy. Adult flies fed with paraquat caused a marked increase of Atg8 puncta formation in midgut cells. Loss of Atg7 or Atg9 resulted in a marked decrease of paraquat-induced Atg8 puncta formation.

3) Figure 7: Depletion of Atg9 in the adult midgut causes a marked decrease in the TSC2 signal level in immunostaining, but it is not very quantitative. The authors should perform immunoblotting to compare the TSC2 protein level in control and Atg9 mutants, as performed in Figure 6.

We thank the reviewer for this comment. We have performed immunoblotting analysis to show that TSC2 level is markedly decreased in the midgut of Atg9 mutants, compared with controls (new data, Figure 8).

4) It remains unclear whether the increased growth of EC in ATG9 mutants is really biologically important. When the increased EC size in ATG9 mutants is rescued (e.g. by TOR inhibition or Patj overexpression in EC cells), are the barrier function and the shortened lifespan restored? These will demonstrate that the increased EC size after a loss of ATG9 is functionally important in affecting the midgut epithelium and lifespan.

We thank the reviewer for this comment. Our new data showed that treating flies with TOR inhibitor rapamycin rescues the intestinal barrier defects of Atg9 mutant animals (new data, Figure 6—figure supplement 1). Moreover, we have previously shown that ablation of Atg9 in EC cells leads to increased lethality in response to paraquat ingestion (Tang et al. 2013, Dev Cell). Strikingly, ectopic expression of Patj in EC cells significantly rescued the paraquat-induced lethality of Atg9 knockdown animals (new data, Figure 7—figure supplement 1). These results together strongly suggest that Atg9 is critical for maintaining midgut epithelium function and integrity.

However, we found that rapamycin treatment could not fully rescue the lifespan defects of Atg9 mutants, compared with controls (new data, Figure 6—figure supplement 1). It is likely that the additional function of Atg9 may contribute to the shortened lifespan in Atg9 mutant animals.

5) Some of the critical data should be quantified (e.g., Figure 1, Figure 1—figure supplement 1, Figure 5, and Figure 6) to strengthen this manuscript.

We have quantified the data in Figure 1, Figure 1—figure supplement 1, Figure 5, and Figure 6.